# An Explorative Analysis of *ABCG2/TOP-1* mRNA Expression as a Biomarker Test for FOLFIRI Treatment in Stage III Colon Cancer Patients: Results from Retrospective Analyses of the PETACC-3 Trial

**DOI:** 10.3390/cancers12040977

**Published:** 2020-04-15

**Authors:** Jan Stenvang, Eva Budinská, Eric van Cutsem, Fred Bosman, Vlad Popovici, Nils Brünner

**Affiliations:** 1Section of Molecular Disease Biology, Institute of Drug Design and Pharmacology, Faculty of Health and Medical Sciences, University of Copenhagen, 2100 Copenhagen, Denmark; stenvang@sund.ku.dk; 2Scandion Oncology, Symbion, 2100 Copenhagen, Denmark; 3RECETOX, Faculty of Science, Masarykova Univerzita, 625 00 Brno, Czech Republic; budinska@recetox.muni.cz; 4Digestive Oncology Department, University Hospitals Leuven and KU Leuven, 3001 Leuven, Belgium; eric.vancutsem@uzleuven.be; 5University Institute of Pathology, University of Lausanne, 1011 Lausanne, Switzerland; Fred.Bosman@chuv.ch

**Keywords:** biomarkers, ABCG2, TOP-1, adjuvant irinotecan, colon cancer

## Abstract

Biomarker-guided treatment for patients with colon cancer is needed. We tested ABCG2 and topoisomerase 1 (TOP1) mRNA expression as predictive biomarkers for irinotecan benefit in the PETACC-3 patient cohort. The present study included 580 patients with mRNA expression data from Stage III colon cancer samples from the PETACC-3 study, which randomized the patients to Fluorouracil/leucovorin (5FUL) +/− irinotecan. The primary end-points were recurrence free survival (RFS) and overall survival (OS). Patients were divided into one group with high ABCG2 expression (above median) and low TOP-1 expression (below 75 percentile) (“resistant”) (*n* = 216) and another group including all other combinations of these two genes (“sensitive”) (*n* = 364). The rationale for the cut-offs were based on the distribution of expression levels in the PETACC-3 Stage II set of patients, where ABCG2 was unimodal and TOP1 was bimodal with a high expression level mode in the top quarter of the patients. Cox proportional hazards regression was used to estimate the hazard ratios and the association between variables and end-points and log-rank tests to assess the statistical significance of differences in survival between groups. Kaplan-Meier estimates of the survival functions were used for visualization and estimation of survival rates at specific time points. Significant differences were found for both RFS (Hazard ratio (HR): 0.63 (0.44–0.92); *p* = 0.016) and OS (HR: 0.60 (0.39–0.93); *p* = 0.02) between the two biomarker groups when the patients received FOLFIRI (5FUL+irinotecan). Considering only the Microsatellite Stable (MSS) and Microsatellite Instability-Low (MSI-L) patients (*n* = 470), the differences were even more pronounced. In contrast, no significant differences were observed between the groups when patients received 5FUL alone. This study shows that the combination of ABCG2 and TOP1 gene expression significantly divided the Stage III colon cancer patients into two groups regarding benefit from adjuvant treatment with FOLFIRI but not 5FUL.

## 1. Introduction

Based on the results from the MOSAIC prospective randomized clinical trial (PRCT) [1], treatment of high risk Stage II and of Stage III colon cancer (CC) patients currently consists of 5-Fluorouracil or Xeloda plus leucovorin (5FUL) plus oxaliplatin (FOLFOX or XELOX). 

Presently, irinotecan is not used in adjuvant treatment of primary CC but only in the metastatic setting [2]. The reason for this is that none of two high-powered and independent PRCTs (PETACC-3 [3] and CALGB 89803 [4]), including high risk Stage II and Stage III colon cancer and randomizing patients to 5FUL ± irinotecan, could demonstrate a significant difference between the treatment groups with respect to recurrence free survival (RFS) or overall survival (OS). 

With a five-year recurrence rate of approximately 30% following adjuvant FOLFOX/XELOX treatment of Stage III CC patients [1], there is obviously a need for other adjuvant treatment modalities using drugs with different mechanisms of action than FOLFOX/XELOX. These treatments must be accompanied by predictive biomarkers, allowing rational allocation of individual patients to the most effective regimen.

Although PRCTs have not shown any added benefit from adjuvant irinotecan of CC cancer when co-administered with 5FUL, it is conceivable that specific subgroups of patients may benefit from the addition of irinotecan treatment but that these patients are concealed within the total patient population. Moreover, given the fundamental differences in molecular mechanisms of action of oxaliplatin and irinotecan as well as the different molecular mechanisms underlying resistance to these two drugs [5], it is likely that the groups of patients benefitting from adjuvant FOLFOX/XELOX or FOLFIRI treatments, respectively, are only partly overlapping, if at all.

We recently characterized isogeneic pairs of CRC cell lines selected for resistance to SN38 (the active metabolite of irinotecan) or oxaliplatin [5]. We identified several genetic aberrations associated with SN38 resistance; in particular, the xenobiotic drug transporter ABCG2 was found to be the most up-regulated gene in the SN38 resistant cell lines [5]. Subsequent functional analyses of this gene demonstrated its major role in SN38 resistance [5]. Moreover, downregulation of the irinotecan target, the topoisomerase-1 enzyme (Top-1), has also been observed in our SN38 resistant cancer cells [6]. Of specific interest is that the resistance mechanisms in our three oxaliplatin-resistant colorectal cancer cell lines [5] did not include regulation of ABCG2 or TOP1 mRNA. In recent publications [5,7], we correlated each of ABCG2 and TOP1 mRNA expression to patient outcome in a subset of Stage III colon cancer patients enrolled in the PETACC-3 study. A trend was demonstrated towards high ABCG2 mRNA expression being correlated with shorter recurrence-free survival (RFS) and shorter overall survival (OS) when compared to patients with low ABCG2 mRNA expression [5], and high TOP1 expression was significantly associated with longer OS but not RFS in FOLFIRI treated patients [7]. We now hypothesize that low TOP1 and high ABCG2 expression (“resistant patients”) define patients who will not benefit from irinotecan containing adjuvant chemotherapy while any other combination of these two genes defines patients (“sensitive patients”) who will benefit from the addition of irinotecan to 5FUL. The present study was designed to test this hypothesis.

## 2. Results

### 2.1. Patient Characteristics

For a detailed description, including a CONSORT diagram on the selection of the present PETACC-3 cohort, please see [7]. Table 1 shows the clinicopathological characteristics of the *n* = 580 stage III CC patients included in the study. For comparison, the clinicopathological characteristics of the full set of 2315 patients from the PETACC-3 Stage III CC patient cohort are included. With gender composition as exception (the subpopulation is slightly enriched in males), the present study population was representative of the global PETACC-3 study population.

Table 2 shows the correlations between clinicopathological parameters and ABCG2 gene expression and TOP1 gene expression, respectively. TOP1 was associated with site, grade, mucinous histology and KRAS mutational status, while ABCG2 was associated with site, MSI and BRAF mutational status. These observations suggest a possible association of ABCG2 with BRAF mutated pathway, and of TOP1 with BRAF-mutant-like (*t*-test *p*-value < 0.001) [8]. In univariate analysis including all 580 patients stratified by treatment arm, no significant benefit from irinotecan addition was found either for RFS or for OS (Figure 1) and thus the selected subgroup does not differ from the main PETACC-3 population with regard to treatment effect.

### 2.2. Combining TOP1 and ABCG2 mRNA Expression

The Spearman’s correlation coefficient between ABCG2 and TOP1 gene expression was r = 0.046 (Appendix A). There were 216 patients in the ABCG2 high/TOP1 low (“resistant patients”) and 364 patients in the “sensitive patient” group. When stratifying the whole set of patients (*n* = 580) according to ABCG2/TOP1 status, a significantly better RFS (Hazard Ratio (HR): 0.75; 95% confidence interval CI: 0.58–0.98; *p* = 0.036) was observed in the “sensitive patient” group as compared to the ABCG2 high/TOP1 low “resistant patient” group (Figure 1 and Appendix A; online only). When stratifying each of the two treatment groups according to the proposed test, the separation between the “sensitive” and “resistant” patient groups in terms of RFS was significant in the FOLFIRI arm (HR: 0.63; 95% CI: 0.44–0.92; *p* = 0.016) but not in the 5FUL arm (Figure 1, Figure 2A,B). 

In terms of relative 3- and 5-years RFS, the patients in the “sensitive” group performed better only under FOLFIRI treatment (relative benefit of 18.2% and 19.9% at 3- and 5-years, respectively–Table 3). The complete pairwise comparisons between all combinations of test group (“sensitive” vs. “resistant” patients) and treatment arm (FOLFIRI vs. 5FUL) (six comparisons) did not yield any statistically significant difference, aside from that between “sensitive” and “resistant” patients groups within the FOLFIRI arm (Figure 1 and Appendix A (online only)).

We also pooled all the 5FUL-only treated patients and estimated the 3- and 5-year RFS (3-years RFS: 68.8%; 5-year RFS: 62.2%). The relative benefit in 3-year and 5-year RFS between FOLFIRI-treated “sensitive” patients and all 5FUL alone treated patients were 4.1% and 9.9%, respectively, in favor of FOLFIRI.

Similar analyses were performed for OS as endpoint. In the whole Stage III population, no statistically significant difference was found between “sensitive” and “resistant” patients (Figure 1 and Appendix A (online only)). However, when analyzing by treatment arm, the FOLFIRI-treated patients labeled as “sensitive” by the test had a significantly longer OS (HR: 0.6; 95% CI: 0.35–0.92; *p* = 0.02) (Figure 1 and Figure 2C), while no such difference could be detected in 5FUL only treated patients (Figure 1 and Figure 2D). When combining the 5FUL patients into one group and then comparing this pooled group with each of the two FOLFIRI treated groups, no significant differences in OS were observed (Appendix A; online only). Nevertheless, the “sensitive” patients treated with FOLIFIRI seemed to fare better with 3- and 5-year relative gains of 1.2% and 6%, respectively (Appendix A; online only). The pairwise comparisons of all possible combinations between test group and treatment arm did not reveal any significant difference with the exception of the one between “sensitive” and “resistant” patients treated with FOLFIRI, already discussed above (Figure 1).

### 2.3. ABCG2 and TOP1 in MSS Plus MSI-L Patient Subgroup

Due to the low number of Microsatellite Instable (MSI) tumors, we focused our analyses on the Stage III Microsatellite Stable (MSS) plus Microsatellite Instable-Low (MSI-L) tumors (*n* = 470). MSS and MSI-L patients in “sensitive patients” treated with FOLFIRI had a significant better RFS (HR: 0.57; 95% CI: 0.37–0.85; *p* = 0.006) (Figure 3A) and OS (HR: 0.57, 95% CI: 0.35–0.92; *p* = 0.02) (Figure 3B) than patients in the “resistant” group. Stratifying the 5FUL only treated MSS patients by the ABCG2 and TOP1 test did not result in any significant separation of the patients for RFS or OS (Figure 1). The 5-year RFS for all MSS plus MSI-L patients treated with 5FUL alone was 59.7% (95% CI: 53.7–66.5), while for “sensitive patients” treated with FOLFIRI it was 69.3% (95% CI: 62.1–77.3), resulting in a relative gain of 15.9% in favor of the latter. When also dichotomizing the 5FUL-only treated group with the biomarker test and when comparing to the FOLFIRI arm, it was seen that FOLFIRI treatment of “sensitive patients” resulted in a 7.3% and 14.3% relative gain in 3-year and 5-year RFS in comparison with the equivalent group treated with 5FUL alone, however without reaching statistical significance. When considering all possible pairwise comparisons of groups defined by the test and treatment arm (Appendix A (online only) for RFS and OS, respectively) the only significant differences were between “sensitive” and “resistant” patients treated with FOLFIRI and between FOLFIRI-treated “resistant patients” and 5FUL-treated group 1 patients (Figure 1).

### 2.4. ABCG2/TOP1 Status as Independent Predictor in Multivariable Models

We tested the independence of ABCG2/TOP1 status in multivariable models including tumor site, MSI status, mucinous histology, and BRAF and KRAS mutation status (without interaction terms). In least absolute shrinkage and selection operator (LASSO) [9] penalized regression analyses, ABCG2/TOP1 status was selected as the most important variable for RFS both in the whole population and in the FOLFIRI-treated arm. In 5FUL, none of the tested variables was found to be significant. Similar results were obtained in the MSS subpopulation, with ABCG2/TOP1 status being selected as the most important variable in whole MSS and in MSS FOLFIRI-treated subpopulations, but not in MSS 5FUL. 

In multivariable Cox regression, after including all the variables selected by penalized regression, ABCG2/TOP1 status had a corresponding adjusted HR: 0.75 (95% CI: 0.57–1.00, *p* = 0.052) for the whole population and HR: 0.72 (95% CI: 0.53–0.96, *p* = 0.028) for the MSS subpopulation. In FOLFIRI arm, ABCG2/TOP1 status had a corresponding HR: 0.63 (95% CI: 0.42–0.94, *p* = 0.020) for all patients and HR: 0.57 (95% CI: 0.38–0.87, *p* = 0.009) for the MSS subpopulation. See Appendix A—Multivariable Regression section.

## 3. Discussion

Irinotecan is a topoisomerase 1 poison and by binding to the Top1 enzyme, toxic complexes are formed leading to induction of apoptosis. We therefore hypothesized that a higher Top1 level in cancer cells is associated with more toxic effects of irinotecan. ABCG2 is a xenobiotic drug efflux pump being involved in outwards transportation of SN38 from cells. An additional hypothesis therefore is that a high cellular level of ABCG2 is associated with less cytotoxic effects of irinotecan.

In a previous study, which included the 580 PETACC-3 patients [5], we found in Stage III CC patients a trend towards association between high ABCG2 expression and poor patient outcome in the irinotecan containing treatment group, but not in the 5FUL only treated group. In another retrospective PETACC-3 study [7], we reported for low TOP1 expression a trend towards an association with short RFS and a borderline significant association with shorter OS in the irinotecan treated patients but not in the 5FUL treated patients [7]. On the assumption that more than one molecular resistance mechanism is involved in irinotecan resistance [5], we now combined ABCG2 and TOP1 expression status in a single dichotomous parameter and hypothesized that patients with a tumor with a high ABCG2 and a low TOP1 expression level might represent those with a low response to irinotecan added to adjuvant 5FUL treatment. As presented in Figure 2, our data confirm this hypothesis in showing that ABCG2/TOP1 status is significantly associated with RFS and OS in Stage III CC patients receiving adjuvant irinotecan containing chemotherapy. In contrast, ABCG2/TOP1 status was neither associated with RFS nor with OS in patients receiving 5FUL only as adjuvant treatment, which is consistent with a predictive rather than a prognostic value.

We compared our results with those published from the MOSAIC study [1], in which 2216 stage III CC patients were randomly assigned to receive 5FUL alone or in combination with oxaliplatin (FOLFOX) for six months. A significant difference (*p* = 0.003) in disease-free survival (DFS) in favor of FOLFOX with a 5-year 8.8% relative increase in DFS in the FOLFOX treated patients was noted and adjuvant FOLFOX is now the standard of care in Stage III CC patients. When we compared 5-year RFS between FOLFIRI “sensitive patients” and the total 5FUL only treated group in our study, we noted that RFS in FOLFIRI treated FOLFIRI “sensitive patients” was 9.9% higher than that of all patients treated with 5FUL only. Thus, the benefit from adjuvant systemic treatment in the MOSAIC study and in the PETACC-3 subgroup of patients with FOLFIRI “sensitive” tumors were comparable at 5 years (the only difference between DFS in the MOSAIC study and RFS in the PETACC-3 study was the inclusion of a second malignancy in the RFS). 

We also performed subgroup analyses, including only MSS + MSI-L patients, which led to even more significant results than those reported for the whole cohort (Figure 3). Klingbiel et al. [10] previously reported that in the PETACC-3 study, MSI status had no effect on survival of FOLFIRI treated patients, neither on RFS nor on OS. Moreover, an interaction test between treatment and MSI status in Stage III patients was not significant. When we included only MSS+MSI-L patients, the differences in RFS and OS between FOLFIRI and 5FUL patients and ABCG2/TOP1 status became more pronounced but still did not reach statistical significance, most probably due to the low number of included patients. 

An interesting point is that the function of ABCG2 can be inhibited in patients [11]. In Scandion Oncology, we develop novel drugs to inhibit ABCG2 [12,13]. When these drugs have been tested in regular clinical phase II trials in patients with metastatic and irinotecan resistant colorectal cancer, they can be taken into randomized clinical testing including Stage III colon cancer patients with high ABCG2 expression. We recently analytically validated commercial antibodies for immunohistochemical (IHC) staining of ABCG2 on formalin-fixed formalin embedded colorectal cancer tissue and identified the BXP21 antibody to fulfill requirements for ABCG2 IHC [14]. 

The major strength of our study lies in the design. In the PETACC-3 PRCT, 5FUL treatment constituted the backbone and irinotecan was added to half of the patients only. This design lends itself to study biomarkers predictive of irinotecan response and to separate a potential predictive from a prognostic impact [15]. Moreover, RFS and OS are valid endpoints for estimating the effect of predictive biomarkers. Finally, the choice of ABCG2 and TOP1 as potential biomarkers for irinotecan sensitivity/resistance was based on a hypothesis derived from results of our in vitro studies on cell lines [5,6]. 

The weakness of the study is related to the lack of an independent validation cohort. However, only one other PRCT (CALGB 89803) has investigated the impact of adding irinotecan to 5FUL in the adjuvant treatment of Stage III CC [4]. Unfortunately, no mRNA expression data are available from the CALGB 89803 study. 

## 4. Materials and Methods 

### 4.1. Patients

The set of patients considered for the present study consisted of all Stage III patients with good quality mRNA expression data (*n* = 580) from the PETACC-3 study [3]. For further information on patient characteristics, inclusion and exclusion criteria, treatment schedules and follow-up, please see the original publication [3]. *n* = 279 of these patients had been randomized to receive adjuvant 5FUL only and *n* = 301 patients received irinotecan in addition to 5FUL. Further details on the present study population are given in Table 1.

All patients signed an informed consent form, allowing collection of tumor tissue for future translational research. Approval for the present translational study was obtained from the PETACC-3 Translational Research Working Party. 

### 4.2. Gene Expression Analyses

As previously described [16], total RNA was extracted from formalin fixed paraffin embedded (FFPE) blocks of the primary cancers. The RNA was amplified and hybridized to the Almac Colorectal Cancer DSA microarray platform (Almac, Craigavon, UK). Whole-genome gene expression data is publicly available from ArrayExpress under accession number E-MTAB-990 [16].

### 4.3. Statistical Methods

The present study was prospective-retrospective in nature. The statistical plan and the applied cutoff values were defined prior to the study. We used the original PETACC-3 study endpoints being RFS and OS. RFS was defined as time in months from randomization until occurrence of local, regional or distant relapse, a second primary colon cancer or death. OS was defined as time in months from randomization until death. For ABCG2, the median from the whole expression data set (Stage II and III) was chosen to dichotomize the patients into ABCG2 high and ABCG2 low. For TOP1 mRNA, we used, based on our previous study [7], the third quartile on the whole expression data of stage II and III patients to group the patients into TOP1 high and low. ABCG2 and TOP1 were then combined into a binary variable: ABCG2 high (above median) and TOP1 low (below the 75-percentile) formed the “resistant patients”, while all other combinations of ABCG2 and TOP1 formed the “sensitive patients “. The patients were further stratified according to the treatment (5FUL or FOLFIRI). 

The Kaplan–Meier method was used to estimate RFS and OS rates, and univariate comparisons were made using the log rank test. The effect size of ABCG2/TOP1 status and treatment arm were estimated in univariate and multivariable analysis using the Cox proportional hazards model. Adjustment variables for multivariable analysis were selected based on LASSO penalized proportional hazards regression [9]. Microsatellite instability (MSI) data were available from a previous study [10] and were tested alongside the clinical and pathological baseline variables: N stage, tumor localization, tumor grade, sex, and age. Formal tests for statistical interaction between dichotomized ABCG2/TOP1 status (“resistant patients” vs “sensitive patients”) and treatment were performed in separate Cox models, including main effects and an interaction term. All results were summarized in terms of hazard ratios (HR), estimated 95% confidence intervals (CI), and *p*-values from the Wald-test.

Pearson correlation coefficients (r) were calculated to test for statistical dependence between the ABCG2/TOP1 variables.

All *p*-values were two-sided and the significance level was set at 0.05. All analyses were performed in R software for statistical computing version 3.4.0 [17].

### 4.4. Subgroup Analyses

Since mechanisms of drug resistance effective in MSI tumors might be different from those in MSS tumors [18], we also divided the Stage III patients into MSS and MSI genotypes, (*n* = 470 tumors being MSS and MSI-L and *n* = 51 tumors being MSI-H (59 missing values)). Kaplan Meier survival statistics were used to estimate RFS and OS rates in each group according to ABCG2/TOP1 status dichotomized as described above.

The REMARK guidelines [19] were followed wherever applicable. 

## 5. Conclusions

In conclusion, we show that ABCG2/TOP1 status as a combined test results is a potential biomarker, which provides significant predictive information on benefit of adjuvant irinotecan treatment of Stage III CC patients. However, our data could only show a trend for a better patient outcome with FOLFIRI treatment of “sensitive patients” as compared to the 5FUL treated patients. The predictive value of our biomarker test needs to be confirmed in an independent validation cohort. Our results also raise the question whether FOLFIRI biomarker positive patients will benefit from FOLFIRI only or whether they are those benefiting from adjuvant treatment with FOLFOX as well. An adjuvant study enrolling Stage III CC patients with a FOLFIRI “sensitive” gene profile and randomizing these patients to treatment with FOLFOX or FOLFIRI will answer this question. 

## Figures and Tables

**Figure 1 cancers-12-00977-f001:**
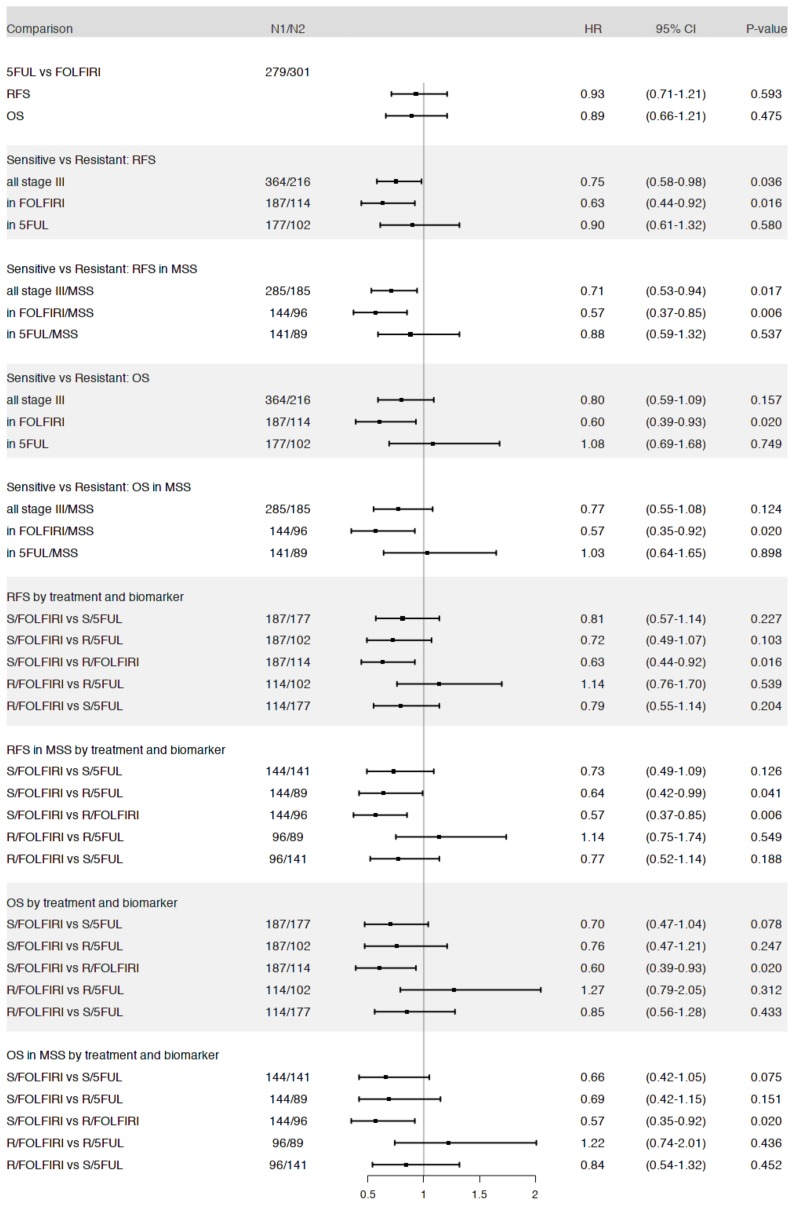
Summary of main survival analysis results: log-rank tests were used to assess the statistical significance of survival differences between groups of interest. Each section of results refers to a set of related tests. The differences assessed are given in the first column, the corresponding sample sizes in the second column (N1/N2: sample size of the first and second group, respectively), while the columns 3–6 summarize the test results in terms of hazard ratios and 95% confidence intervals (plots in column 3) and corresponding *p*-values (log-rank test—column 6). In the first column, “sensitive” was abbreviated as “S” and “resistant” as “R”, respectively. Thus, “S/FOLFIRI” stands for “sensitive under FOLFIRI treatment” etc.

**Figure 2 cancers-12-00977-f002:**
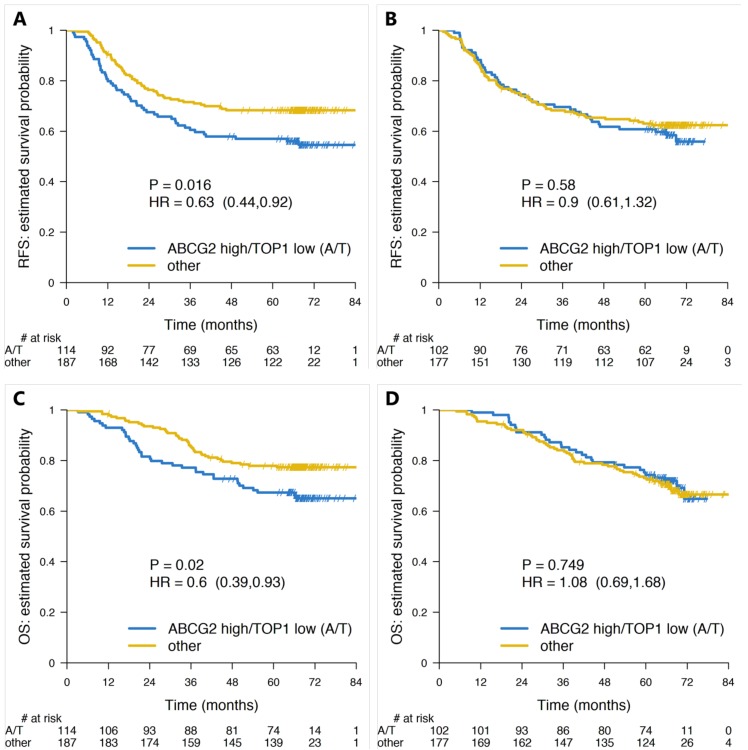
Survival plots (Kaplan-Meier estimates) for “resistant” (ABCG2-high/TOP1-low, abbreviated A/T under the plots) and “sensitive” (all other combinations of ABCG2 and TOP1 genes) patient groups in whole Stage III cohort (*n* = 580). The four plots show the RFS of “resistant” (blue line) and “sensitive” (gold line) under (**A**) Fluorouracil/leucovorin (5FUL) + irinotecan (FOLFIRI) and (**B**) 5FUL treatments and the overall survival (OS) of the same groups under (**C**) FOLFIRI and (**D**) 5FUL treatments, respectively. Numbers at risk are given under each plot.

**Figure 3 cancers-12-00977-f003:**
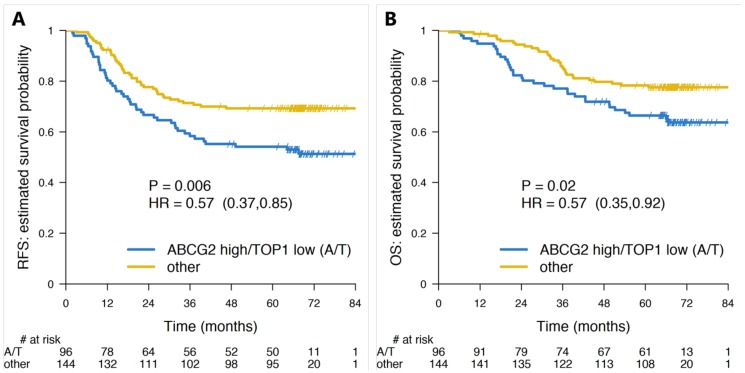
Survival plots (Kaplan-Meier estimates) for “resistant” (ABCG2-high/TOP1-low, abbreviated A/T under the plots) and “sensitive” (all other combinations of ABCG2 and TOP1 genes) patient groups in stage III/MSS (Microsatellite Stable) subset (*n* = 470). The two plots show the (**A**) Recurrence free survival (RFS) and (**B**) Overall Survival (OS) of “resistant” (blue line) and “sensitive” (gold line) under FOLFIRI treatment, respectively. Numbers at risk are given under each plot.

**Table 1 cancers-12-00977-t001:** Population characteristics for the whole PETACC-3/Stage III and the study subpopulation. The only statistically significant difference was between male/female proportions (* starred covariate in the table; *p* = 0.025). The missing values (denoted NA (not available)) were not considered when computing the proportions. Microsatellite Instability (MSI) Status is divided into MSI High (MSI-H), MSI Low (MSI-L) and Microsatellite Stable (MSS).

Variables	All PETACC-3 Stage III (*n* = 2315)	Study Subpopulation (*n* = 580)
**Age (mean (sd))**	58.35 (10.54)	58.86 (10.44)
**Sex * (n (%))**		
Male	1263 (54.6)	347 (59.8)
Female	1052 (45.4)	233 (40.2)
**Treatment (n (%))**		
5FUL	1157 (50.0)	279 (48.1)
FOLFIRI	1158 (50.0)	301 (51.9)
**Site (n (%))**		
left	1422 (61.4)	366 (63.1)
right	893 (38.6)	214 (36.9)
**Grade (n (%))**		
1,2	877 (88.1)	512 (88.9)
3,4	119 (11.9)	64 (11.1)
NA	1319	4
**T-stage (n (%))**		
T1, T2	196 (8.5)	51 (8.8)
T3	1766 (76.4)	438 (75.5)
T4	351 (15.2)	91 (15.7)
NA	2	0
**N-stage (n (%))**		
N0, N1	1496 (64.4)	377 (65.0)
N2	819 (35.4)	203 (35.0)
**Mucinous histology (n (%))**		
No	807 (81.0)	477 (82.8)
Yes	189 (19.0)	99 (17.2)
NA	1319	4
**MSI status (n (%))**		
MSI-H	106 (12.1)	51 (9.8)
MSI-L, MSS	772 (87.9)	470 (90.2)
NA	1437	59
**BRAF V600E (n (%))**		
mutated	78 (8.4)	37 (6.7)
wild type	848 (91.6)	512 (93.3)
NA		31
**KRAS codon 12, 13 (n (%))**		
mutated	364 (39.6)	219 (40.1)
wild type	556 (60.4)	327 (59.9)
NA	1395	34

**Table 2 cancers-12-00977-t002:** Comparison of expression levels between various stratifications in the study subpopulation for ABCG2 and TOP1 genes, respectively. For each gene, the mean and standard deviation of the expression levels (log2) are indicated and the corresponding p-values from Student’s *t*-test (significant are emphasized by italic) for binary categories and ANOVA for multiple categories.

Stratification Factor	n (%)	TOP1 (Mean (sd))	ABCG2 (Mean (sd))
**Site**			
left	366 (63.1)	4.85 (1.00)	2.43 (0.55)
right	214 (36.9)	4.59 (0.98)	2.55 (0.73)
*p*-value		*0.002*	*0.025*
**Grade**			
1, 2	512 (88.9)	4.79 (0.97)	2.46(0.59)
3,4	64 (11.1)	4.49 (1.21)	2.59 (0.85)
*p*-value		*0.026*	0.120
**T-stage**			
T1, T2	51 (8.8)	4.86 (1.10)	2.47 (0.46)
T3	438 (75.5)	4.75 (0.96)	2.48 (0.65)
T4	91 (15.7)	4.74 (1.12)	2.46 (0.55)
*p*-value		0.725	0.961
**N-stage**			
N0, N1	377 (65.0)	4.80 (0.95)	2.44 (0.52)
N2	203 (35.0)	4.68 (1.08)	2.54 (0.78)
*p*-value		0.162	0.079
**Mucinous histology**			
no	477 (82.8)	4.85 (0.98)	2.48 (0.63)
yes	99 (17.2)	4.32 (0.97)	2.43 (0.61)
*p*-value		*< 0.001*	0.411
**MSI status**			
MSI-H	51 (9.8)	4.52 (1.01)	2.20 (0.40)
MSI-L, MSS	470 (90.2)	4.78 (0.99)	2.49 (0.59)
p-value		0.074	*0.001*
**BRAF V600E mutation**			
mutated	37 (6.7)	4.55 (1.15)	2.81 (0.98)
wild type	512 (93.3)	4.77 (0.99)	2.44 (0.53)
*p*-value		0.202	*< 0.001*
**KRAS codon 12, 13**			
mutated	219 (40.1)	4.65 (0.97)	2.45 (0.55)
wild type	327 (59.9)	4.82 (1.01)	2.48 (0.59)
*p*-value		*0.040*	0.549

**Table 3 cancers-12-00977-t003:** Summary of patient survival rates by treatment and biomarker group (R: resistant, S: sensitive) at 3 and 5 years, respectively. The relative benefit is denoted by (S−R)/R.

**End-point**	**FOLFIRI**							
	**S vs R**		**3-year survival rates**			**5-year survival rates**		
	**HR (95% CI)**	***p*-value**	**S (%) (95% CI)**	**R (%) (95% CI)**	**(S−R)/R (%)**	**S (%) (95% CI)**	**R (%) (95% CI)**	**(S−R)/R (%)**
**RFS**	0.63 (0.44–0.92)	*0.016*	71.5 (65.3–78.3)	60.5 (52.2–70.5)	18.2	68.3 (61.9–75.3)	57.0 (48.6–66.8)	19.9
**OS**	0.60 (0.39–0.93)	*0.020*	85.5 (80.6–90.7)	77.2 (69.9–85.3)	10.8	77.9 (72.2–84.1)	67.3 (59.2–76.6)	15.7
	**5FUL**							
	**S vs R**		**3-year survival rates**			**5-year survival rates**		
	**HR (95% CI)**	***p*-value**	**S (%) (95% CI)**	**R (%) (95% CI)**	**(S−R)/R (%)**	**S (%) (95% CI)**	**R (%) (95% CI)**	**(S−R)/R (%)**
**RFS**	0.90 (0.61–1.32)	0.58	68.3 (61.7–75.5)	69.6 (61.2–79.1)	−0.02	63.0 (56.3–70.6)	60.8 (52.0–71.0)	0.04
**OS**	1.08 (0.69–1.68)	0.75	84.1 (78.8–89.7)	85.2 (78.6–92.4)	−0.01	73.1 (66.8–80.0)	74.3 (66.2–83.3)	−0.02

Abbreviations: HR (Hazard Ratio), CI (Confidence Interval), FOLFIRI (Fluorouracil/leucovorin (5FUL) + irinotecan), RFS (Recurrence Free Survival), OS (Overall Survival).

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
