# Peer review of "An Explorative Analysis of *ABCG2/TOP-1* mRNA Expression as a Biomarker Test for FOLFIRI Treatment in Stage III Colon Cancer Patients: Results from Retrospective Analyses of the PETACC-3 Trial"

_cancers, 2020, doi:10.3390/cancers12040977_

Round 1

Reviewer 1 Report

The authors used two prior published data (ABCG2, reference 5; TOP1, reference 7) and combined into a predictor ("resistant", high ABCG2 [above median] plus low TOP1 [below 75 percentile]) of irinotecan treatment for stage III colon cancer. The statistics is convincing because the predictor was only significant in patients who accepted the treatment of irinotecan. There is some minor spell check required.

Line 33, what is MSL

Line 45, what is PRCT

Line 357, no detailed reference

Author Response

We thank Reviewer 1 for the very positive comments and appreciate the good suggestions to further improve the manuscript.

The authors used two prior published data (ABCG2, reference 5; TOP1, reference 7) and combined into a predictor ("resistant", high ABCG2 [above median] plus low TOP1 [below 75 percentile]) of irinotecan treatment for stage III colon cancer. The statistics is convincing because the predictor was only significant in patients who accepted the treatment of irinotecan. There is some minor spell check required.

Line 33, what is MSL

Response: In Petacc3 the applied terminology regarding MSS status is the following: MSI-H (MSI high), MSI-L (MSI low) and MSS. In all analyses, we considered MSI-L and MSS together - and generally call them MSS. The MSI-H group was simply named MSI.

The MSL in line 33 is therefore changed to MSI-L.

In Table 1 “MSIL” has been changed to “MSI-L”

Line 45, what is PRCT

Response: PRCT is an abbreviation for “prospective randomized clinical trial”. This is defined in line 41 (the first line in the introduction). No changed has been made.

Line 357, no detailed reference

Response: This has been corrected

Reviewer 2 Report

The topic of the manuscript is very interesting, since it explores possible biomarker-guided treatment for patients with colon cancer. In particular, in the large PETACC-3 patient cohort, the authors explore the hypothesis that low TOP1 and high ABCG2 expression (resistant patients) defines patients who will not benefit from irinotecan containing adjuvant chemotherapy, in contrast to any other combination of the expression levels of the two genes, defining patients (sensitive patients) that will benefit from the addition of irinotecan to 5FUL. 

I have just minor suggestions regarding the abstract:

I am aware that it is hard to summary the complex anlyses of the manuscript, however I would suggest to include a simple statement about the rationale of dividing patients in the two main groups, i.e. high ABCG2 and low TOP-1 expression, and all other expression levels combinations. The reader will find exhaustive explanation for that in "Introduction" (your in vitro study) and also in "Discussion" (your data on patients), however it could be helpful to have this notion in the abstract for the non-insiders.

Related to that, I also suggest to specify all the abbreviations, e.g. Top-1 (topoisomerase 1), FOLFIRI ( ) etc. 

Author Response

Reviewer 2:

We thank Reviewer 2 for the very positive comments and appreciate the good suggestions to further improve the manuscript.

I am aware that it is hard to summary the complex anlyses of the manuscript, however I would suggest to include a simple statement about the rationale of dividing patients in the two main groups, i.e. high ABCG2 and low TOP-1 expression, and all other expression levels combinations. The reader will find exhaustive explanation for that in "Introduction" (your in vitro study) and also in "Discussion" (your data on patients), however it could be helpful to have this notion in the abstract for the non-insiders.

Response:

We are pleased that the reviewer are satisfied with the explanations about the subgrouping and analyses in the main manuscript.

However, we agree that a short statement in the abstract would be helpful for the non-insiders.

Therefore, a sentence has now been included in the abstract to clarify the rationale for the cut-off values defining the groups.

Line 27-30: “The rationale for the cut-offs were based on the distribution of expression levels in the PETACC-3 stage II set of patients, where ABCG2 was unimodal and TOP1 was bimodal with a high expression level mode in the top quarter of the patients.”

For the reviewer we have here included a slightly more elaborate explanation.

To address the questions of dividing in to two groups and the rationale for the chaosen cut-off values:

(i) why split the population in 2 groups?

It was necessary to identify a potential set of responder patients, hence we clearly specified what is meant by "high ABCG2" and "low TOP-1" for the sensitive group. The resistant group was consequently defined as "all others".

(ii) why those cut-offs (median for ABCG2 and 75 percentile for TOP-1)?

We inspected the distribution of the expression levels of the two genes in the stage II set of patients of PETACC3 (disjoint of the set considered in the article) and we found that ABCG2 is unimodal - hence we chose a conservative, generally accepted, threshold (the median) - and that TOP-1 seemed to be bimodal with a "high expression level" mode in the top third/top quarter of the patients - hence we defined "high expression" as the top 25 percentile. This is illustrated in the plot below. We did not have enough stage III patients to perform threshold optimization and independent validation, hence we proceeded with the cut-offs defined as above fixed beforehand.

Gene expression distribution in the stage II patients in the PETACC-3 cohort: Red: ABCG” distribution; Green: TOP1 distribution.

Related to that, I also suggest to specify all the abbreviations, e.g. Top-1 (topoisomerase 1), FOLFIRI ( ) etc.

Response:

We have specified the relevant abbreviations in the abstract. Line: 21, 24, 36
